# A Robust Model Predictive Control for a Photovoltaic Pumping System Subject to Actuator Saturation Nonlinearity

**Omar Hazil** [1], **Fouad Allouani** [2], **Sofiane Bououden** [2,*], **Mohammed Chadli** [3], **Mohamed Chemachema** [4], **Ilyes Boulkaibet** [5] and **Bilel Neji** [5]

1   Centre de Développement des Energies Renouvelables, Algiers 16340, Algeria
2   Laboratory of SATIT, Department of Industrial Engineering, Abbes Laghrour University, Khenchela 40004, Algeria
3   IBISC, Université Paris-Saclay, Univ Evry, 91020 Evry, France
4   Department of Electronics, Faculty of Technology, University of Constantine 1, Campus A. Hamani, Route Ain El Bey, Constantine 25017, Algeria
5   College of Engineering and Technology, American University of the Middle East, Egaila 54200, Kuwait
*   Correspondence: sofiane.bououden@univ-khenchela.dz

**Abstract:** In this paper, a new robust model predictive control (RMPC) for uncertain nonlinear systems subject to actuator saturation is designed to regulate the terminal voltage of a photovoltaic generator (PVG) that feeds a DC motor-pump via a buck DC–DC converter. The considered system is a combination of a PVG-converter and DC motor-pump, which possesses nonlinear behavior along with under a saturating control signal highly dependent on the operation point and climate conditions of solar radiation and temperature. As a result, the control task is complex due to the nonlinearity of the system and its dependence on climate conditions. Based on the dead-zone property, the presented paper introduces a new RMPC technique to provide an innovative and efficient solution to ensure the closed-loop system's robust stability in the presence of actuator nonlinearity. In this paper, the nonlinear system is described in polytypic form, and an appropriate linear feedback control law is designed and used to minimize an infinite horizon cost function under the framework of linear matrix inequalities (LMIs). Furthermore, sufficient state-feedback control law conditions are synthesized to guarantee the robust stability of the closed-loop system in the presence of polytypic uncertainties. Simulation results are provided, in which the results illustrate the effectiveness of the proposed method.

**Keywords:** photovoltaic pumping system; DC–DC buck converter; robust model predictive control; nonlinear system; actuator saturation; linear matrix inequalities; polytypic system

## 1. Introduction

Over the past three decades, solar energy has become increasingly popular as one of the main renewable energy sources. Several strategies have been introduced to use sunlight as a source of energy in which sunlight can be converted into heat (solar–thermal energy conversion) [1], electricity (solar–photovoltaic energy conversion) [2], solar fuel (hydrogen) generated via photocatalytic water splitting [3,4], or sunlight chemicals, such as $H_2O_2$ production [5]. It can also be used for $CO_2$ reduction [6] and ammonia synthesis [7]. The use of photovoltaic energy has grown tremendously in a wide range of applications, where many non-electrified villages and rural regions rely on solar energy as their primary source of electricity since electrification is difficult and expensive [8,9]. In addition, photovoltaic water-pumping systems for irrigation and water supply in remote areas have been widely implemented due to their unique features of ease of installation, environment friendliness, and low maintenance costs [10,11]. For the optimization of energy, PV water-pumping systems have to operate at their maximum power point (MPP). A directly coupled PV electromechanical system operates at the intersection of current–voltage curves of the PV

array and DC motor-pump set [12,13]. PV water-pumping systems are usually designed to extract as much energy as possible from the solar resource, using a simple DC–DC power converter controlled by an maximum power point tracker (MPPT) algorithm [14,15]. In the literature, many of the MPPT algorithms have been applied for obtaining the maximal power from PV systems. Perturb and observe (P&O) and incremental conductance (INC) are the most widely used methods [16–18]. However, the simplest and fastest algorithm is the constant voltage control approach, in which the PV array is controlled to operate at a constant voltage equal to the MPP voltage of the array at the standard test condition (STC) provided by the manufacturer, by adjusting the duty ratio of the power converter [15]. The authors in [19,20] have shown that this technique has improved stability and low dependence on solar irradiation. Elgendy et al. [13] proved in a comparative investigation that it offers significantly better energy use efficiencies (up to about 91%) compared to directly connected systems without taking the effects of insolation and temperature variations on the MPP voltage into consideration.

Generally, the dynamic behavior of power converter systems can be described as a bilinear model under a saturating control signal. The classical control techniques applied to power converters usually do not take into account input saturation, which can severely degrade the performance of the closed-loop system, thus making the closed-loop system unstable, especially if the converter is subject to large perturbation. Furthermore, when a power converter is used in solar applications, the control system becomes more complex. Many studies have lately demonstrated that PVG characteristics have a substantial effect on the dynamic behavior of power converters [21]. Since the PVG dynamic resistance is both an environmental variable and operating point dependent, major changes in the PV system might compromise its stability [15].

Despite the nonlinearity and uncertainty of these systems, the linear (state or output) feedback control of power converters based on the small signal control theory proposed by Middlebrook and Cuk [22] is the most applicable approach at present. For example, the fuzzy-logic controller proposed in [23] and the PID controller proposed in [24] are based on small signal control theory. Such linear controllers are designed based on the model linearization at a certain operation point, and the obtained model can only be useful for small variations around that specific steady point [25,26]. To cope with the deficiency of these linear controllers, several nonlinear control strategies have been developed with stability analysis, including fuzzy-logic controllers [27,28], adaptive control [29,30], neural-network-based control [31], sliding mode control (SMC) [32], and feedback linearization controllers [33]. However, these methods have two major limitations: The constraints on the input actuators are not considered when using these methods. In addition, the closed-loop performance in terms of robust stability and the presence of parameter uncertainties is not optimized. In recent years, more consideration has been given to studying intelligent and robust control structures, which can maintain nonlinear systems stability over a certain operating range.

Model predictive control (MPC) is a control strategy that offers attractive solutions for controlling constrained linear or nonlinear systems [34,35]. The ability to handle hard constraints on states/outputs and inputs and time-varying behaviors make the MPC method an effective technique for a wide range of practical applications [36]. MPC algorithms are widely based on dynamic models to predict system behaviors over a prediction horizon. The reliable solution obtained is based on the minimization at each time step of an upper bound of the worst-case infinite horizon quadratic function.

In general, it is difficult to construct an accurate model of a system in which uncertainty is frequently present. It is well known that the uncertainty related to systems can be represented in the form of parametric uncertainty or bounded disturbance regions. In the presence of a significant level of uncertainty, the control law designed by MPC based on a nominal model is suboptimal and can even be infeasible. Unfortunately, traditional MPC approaches fail to explicitly handle the plant-model uncertainty. As a result, numerous researchers have recently shown great interest in the RMPC of uncertain systems, where

several tools, such as the LMI optimization approach, were developed to build RMPC algorithms [37–40].

In most practical control processes, actuator saturation is a common kind of nonlinearity. The classical control techniques that ignore actuator saturation may severely reduce the performance of a closed-loop system, and the closed-loop system may lose its stability, especially if subjected to certain large perturbations [41,42]. Hence, it is important to consider the control input constraints in the RMPC design. In the literature, stability analysis and the synthesis of RMPC strategies for linear systems with actuator saturation have been considered as a popular topic in the past few decades. The common feature of most RMPC strategies is that they deal with input nonlinearity by estimating the domain of attraction in the presence of actuator saturation. Kothare et al. [37] proposed a new robust centralized constrained MPC technique using an LMI-based optimization approach to guarantee the asymptotic stability of closed-loop systems. The proposed algorithm allows the incorporation of a large class of plant uncertainty descriptions and can guarantee the stability and robustness of the controlled system. The design in [37] also includes systematic treatment of input and state constraints. Casavola et al. [38] presented a scheduling min-max MPC algorithm for linear parameter varying (LPV) systems with polytopic uncertainty subject to input saturation. Cao and Lin [43] proposed an RMPC algorithm for LPV systems with polytopic uncertainty in the presence of actuator saturation. In the approach in [43], the actuator saturation constraint was characterized in terms of the convex hull of a group of auxiliary linear feedback laws and an actual linear feedback law. Huanga et al. [44] improved the design in [43] by taking the relative weighting between the auxiliary and actual feedback laws into account. Despite the improvements obtained, this method does not consider certain issues of system nonlinearity. Another disadvantage is that the real-time implementation of this approach suffers from a heavy computational burden due to the large number of inequalities when the saturating linear feedback law is expressed on a convex hull. This may lead to the intractability of the optimization problem, especially in the case of high-dimensional systems and fast sampling applications. Moreover, numerous MPC designs for linear systems with actuator saturation have been published in [45–49].

During the past decade, many MPC techniques for systems subject to actuator saturation have been studied. In [50], a model predictive tracking control algorithm for the flexible air-breathing hypersonic flight vehicle was proposed to handle the flight control problem with actuator constraints and input delays. A novel, fast MPC technique with actuator saturation for large-scale structures based on the explicit expression form of the Newmark-$\beta$ method and parametric variational principle was introduced in [51]. In [52], the asymptotic stability of the finite horizon MPC of nonlinear systems with incremental input constraints was proposed and investigated. The authors in [53] presented an MPC approach for systems with a dead zone and saturation, where two different control strategies based on MPC were compared: the former uses hybrid MPC, while the latter is based on dead-zone inversion and standard MPC. A novel discrete-time sliding mode predictive control technique for a tethered satellite with saturated input was proposed and investigated in [54], in which a discrete-time auxiliary controller was involved in the discrete-time MPC scheme to achieve fast and more stable performance. However, in contrast to linear systems subject to actuator saturation, there are few significant MPC designs for nonlinear systems subject to actuator saturation.

In this work, the ability of the LMI approach is exploited to accommodate an MPC-based technique for the control of nonlinear systems subject to actuator saturation, through the dead-zone property. As the main contribution in this work, a new RMPC for uncertain nonlinear systems subject to actuator saturation is proposed to control a photovoltaic pumping system where a simulation study on the PV pumping system, which comprises a PVG, a voltage-mode-controlled DC–DC buck converter, and a DC motor-pump, is illustrated to evaluate the performance of the proposed controller. The objective of the control process is to keep the PVG voltage at the MPP voltage in the presence of PVG dynamic resistance uncertainty and atmospheric condition changes. The proposed RMPC

algorithm innovatively uses certain ideas from the work in [55]. In the proposed approach, a saturation model based on dead-zone nonlinearity is used to handle the saturation of the control input. The controller design is characterized as an optimization problem of the worst-case performance objective function over an infinite moving horizon. In addition, at each time step, adequate state feedback that renders the closed-loop saturated nonlinear system globally asymptotically stable to the origin is obtained via linear matrix inequalities. The main advantages of the proposed approach, compared with other well-known RMPC techniques, are its ability to consider both actuator saturation and system nonlinearity and the reduction in conservativeness by avoiding a large number of inequalities when the actuator saturation constraint is characterized in terms of the convex hull; therefore, the proposed algorithm guarantees a lower computation time compared to other methods. The proposed method can be applied to deal with complex industrial systems, such as computer numerical control (CNC) machines, active magnetic bearing (AMB), robot manipulators, overhead cranes, and DC–DC power converters. Different scenarios are used in the simulation study to illustrate the effectiveness and applicability of the proposed approach to the PV pumping system.

The rest of this paper is organized as follows: Section 2 describes the mathematical formulation of the problem to be handled. Some basic concepts concerning MPC are introduced and the saturated RMPC scheme is proposed in Sections 3 and 4, respectively. Next, several simulation results are demonstrated under the Matlab environment in Section 5 to test the validity and effectiveness of our method. Finally, the paper is concluded in Section 6.

## 2. Problem Formulation

Consider a nonlinear discrete-time dynamical system subject to actuator saturation:

$$x(k+1) = f(x(k), \overline{u}(k)) \tag{1}$$

where $k$ is the current time instant and $x \in \mathbb{R}^n$ is the state. The control input is bounded as $\overline{u}(k) = sat(u(k))$. $sat(u(k)) \in \mathbb{R}^m$ is a saturation function of the control input $u(k) \in \mathbb{R}^m$, which is defined as:

$$sat(u(k)) = \begin{cases} -u_{lim}, & if & u(k) < -u_{lim} \\ u(k), & if & -u_{lim} \leq u(k) \leq u_{lim} \\ u_{lim} & if & u(k) > u_{lim} \end{cases} \tag{2}$$

where $u_{lim}$ is a control input limit and $u_{lim} = u_{max}$. $u_{max}$ is the maximum control input limit. $f \in \mathbb{C}^2$ is a nonlinear function of states and control inputs $f(0,0) = 0$. Let $A = \frac{\partial f}{\partial x}\big|_{(0,0)}$, $B = \frac{\partial f}{\partial u}\big|_{(0,0)}$, and the dynamic system (Equation (1)) can be reformulated as a polytopic uncertain system:

$$x(k+1) = A(\beta)x(k) + B(\beta)\overline{u}(k) + \widetilde{f}(x(k), \overline{u}(k)) \tag{3}$$

where the system matrices are affine functions of a parameter vector $\beta$ of $r$ parameters ($\beta = (\beta_1, \ldots, \beta_r)$). Each uncertain parameter $\beta_j$ is bounded between a minimum and a maximum value $\overline{\beta_j}$ and $\underline{\beta_j}$. $j = \{1, 2, \ldots, r\}$

$$\beta_j \in \left[\underline{\beta_j}, \overline{\beta_j}\right] \tag{4}$$

Moreover, we assume that:

$$[A(\beta)\ B(\beta)] \in \Omega = Co\{[A_1\ B_1], [A_2\ B_2], \ldots, [A_r\ B_r]\} \tag{5}$$

where $\Omega$ denotes the convex hull and $\begin{bmatrix} A_j & B_j \end{bmatrix}$ are vertices of the convex hull. $\Omega$ can be written as:

$$[A(\beta)\ B(\beta)] = \left\{ \sum_{j=1}^{r} \beta_j [A_j\ B_j],\ \beta_j > 0,\ \sum_{j=1}^{r} \beta_j = 1 \right\} \tag{6}$$

$\widetilde{f}(x(k), \overline{u}(k))$ is the nonlinear term obtained by differing between the nominal nonlinear model and the linear part, which is given by:

$$\widetilde{f}(x(k), \overline{u}(k)) = f(x(k), \overline{u}(k)) - A_j x(k) + B_j \overline{u}(k) \tag{7}$$

Assume that $\widetilde{f}$ is globally Lipschitz or at least locally Lipschitz in a region $\mathfrak{D}$ including the origin with respect to $x(k)$ and uniformly in $\overline{u}(k)$. Therefore, we have the following Lipschitz condition:

$$\left| \left| \widetilde{f}(x, \overline{u}) - \widetilde{f}(x_0, \overline{u}) \right| \right|_2 \leq N ||x - x_0||_2 \tag{8}$$

$$\begin{cases} \forall x, x_0\ \in \mathbb{R}^n \text{ globally Lipschitz} \\ \quad \forall x, x_0\ \in \mathfrak{D} \text{ locally Lipschitz} \end{cases}$$

where $N \in \mathbb{R}^{n \times n}$ is a Lipschitz constant matrix. For $x_0 = 0$, the Lipschitz inequality (Equation (8)) can be rewritten as:

$$\left[ \widetilde{f}(x, \overline{u}) - \widetilde{f}(0, \overline{u}) \right]^T I \left[ \widetilde{f}(x, \overline{u}) - \widetilde{f}(0, \overline{u}) \right] \leq x^T N^T N x \tag{9}$$

where $I$ is the identity matrix.

This paper aims to find a non-saturating linear-state feedback law $u(k) = Hx(k)$, where a dead-zone nonlinearity approach is used to handle the saturation nonlinearity. Hence, the dynamic system (Equation (3)) is written as:

$$\begin{aligned} x(k+1)\ &= A_j x(k) + B_j(\overline{u}(k) - u(k) + u(k)) + \widetilde{f}(x(k), \overline{u}(k)) \\ &= \left( A_j + B_j H \right) x(k) + B_j \psi(k) + \widetilde{f}(x(k), \overline{u}(k)) \end{aligned} \tag{10}$$

where $H$ is the state feedback gain. Let us introduce the following useful lemmas for later use.

**Lemma 1.** *For the saturation constraint defined by Equation (2), let:*

$$\varphi = \overline{u} - u \tag{11}$$

*Thus, there is:*

$$\varphi^T \varphi \leq \epsilon u^T u \tag{12}$$

*where $0 < \epsilon < 1$ and $\psi = [\varphi_1, \varphi_2, \ldots, \varphi_N]^T \in \mathbb{R}^N$. $\varphi_i (i = 1, 2, \ldots, N)$ is the dead-zone nonlinearity function.*

**Proof .** See [56]. $\square$

**Lemma 2.** *Schur complements lemma: for any three matrices functions: $L(x) = L(x)^T$, $M(x) = M(x)^T$, and $W(x)$. The LMI*

$$\begin{bmatrix} L(x) & W(x) \\ W(x)^T & M(x) \end{bmatrix} > 0 \tag{13}$$

*is equivalent to*

$$M(x) > 0,\ L(x) - W(x)M(x)^{-1}W(x)^T > 0 \tag{14}$$

*or*

$$L(x) > 0,\ M(x) - W(x)L(x)^{-1}W(x)^T > 0 \tag{15}$$

**Proof.** Refer to [57]. □

**Lemma 3.** *Assume that F and E are vectors or matrices with appropriate dimensions. For any positive scalar $\alpha > 0$, the following inequality holds:*

$$F^T E + E^T F \leq \alpha F^T F + \alpha^{-1} E^T E \tag{16}$$

## 3. Robust Model-Based Predictive Control Using LMIs

Let $x(k+1|k) \in \overline{\mathbb{X}}$ be the predicted state of the plant at time $k+1, i \geq 0$ and $u(k+1|k) \in \overline{\mathbb{U}}$ the future control move at time $k+1, i \geq 0$, in which $\overline{\mathbb{X}}$ and $\overline{\mathbb{U}}$ are compact subsets of $\mathbb{R}^n$ and $\mathbb{R}^m$, respectively, and both of them contain the origin as an interior point. The RMPC aims to find an efficient state feedback control law $u(k+i|k) = H(k)x(k+i|k)$, $i \geq 0$ by minimizing the following worst-case performance function:

$$J(k) = \sum_{i=0}^{\infty} x(k+i|k)^T S x(k+i|k) + \overline{u}(k+i|k)^T R \overline{u}(k+i|k) \tag{17}$$

where $S \in \mathbb{R}^{n \times n}$ and $R \in \mathbb{R}^{m \times m}$ are positive definite states and control weights, respectively. The optimization problem (Equation (17)) can be formulated as follow:

$$\min_{u(k+i|k)} J(k) \tag{18}$$

Let us introduce a quadratic Lyapunov function $V(x) = x^T P x$, $P > 0$ of the state $x(k|k)$, with $V(0) = 0$, at sampling time $k$. Suppose the following robust stability condition is satisfied:

$$V(k+i+1|k) - V(k+i|k) \leq - \left[ x(k+i|k)^T S x(k+i|k) + \overline{u}(k+i|k) \right)^T R \overline{u}(k+i|k) \right] \tag{19}$$

Summing this inequality from $i = 0$ to $i = \infty$, we get:

$$x(\infty|k)^T P x(\infty|k) - x(k|k)^T P x(k|k) \leq -J \tag{20}$$

With $x(\infty|k) = 0$ or $V(x(\infty|k)) = 0$, the upper bound of the cost function is:

$$J \leq x(k|k)^T P x(k|k) \leq \gamma \tag{21}$$

where $\gamma$ is a positive scalar and is regarded as an upper bound of the objective in Equation (17):

$$\sum_{i=0}^{\infty} x(k+i|k)^T S x(k+i|k) + \overline{u}(k+i|k)^T R \overline{u}(k+i|k) \leq \gamma \tag{22}$$

Thus, the control action of robust MPC at time $k$ can be obtained by solving the following optimization problem:

$$\min_{u(k+i|k)} \gamma \tag{23}$$

Applying Schur complements, the condition $x(k|k)^T P x(k|k) \leq \gamma$ in Equation (21) can be expressed equivalently as the LMI:

$$\begin{bmatrix} I & * \\ x(k) & Q \end{bmatrix} \geq 0, \ Q > 0 \tag{24}$$

We can postulate the following theorems to construct the state feedback matrix $H$ in order to minimize the upper bound $\gamma$. (More details can be found in Kothare et al. [37]).

## 4. Main Results

### 4.1. Linear Systems Subject to Actuator Saturation

In this section, for the case in which the nonlinear term is not considered, a saturated controller is designed to stabilize the input-saturated linear system (Equation (25)):

$$x(k+1) = A_j x(k) + B_j \overline{u}(k), \ j = \{1, 2, \dots, r\} \tag{25}$$

where $\overline{u}(k) \in \mathbb{R}^m$ is the saturation function of the control input defined in Equation (2). With the dead-zone nonlinearity expression of a saturating linear feedback law, as described in Lemma 1, the first result of our work is given by the following theorem.

**Theorem 1.** *Consider the discrete-time linear system (Equation (25)) and assume $x(k|k)$ is the measured state of $x(k)$ at each sample time $k$. The state feedback control law $u(k + i|k) = H(k)x(k + i|k)$, $i \geq 0$ that minimizes the upper bound $\gamma$ on the infinite horizon quadratic performance index $J(k)$ is given by $H(k) = YQ^{-1}$, where $Q$, $Q > 0$, and $Y$ are the solutions to the following LMIs:*

$$\begin{array}{c} min \ \gamma \\ \gamma, \xi_1, \xi_2, Q, Y \\ subject \ to \end{array} \tag{26}$$

$$\begin{bmatrix} I & * \\ x(k) & Q \end{bmatrix} \geq 0 \tag{27}$$

$$\begin{bmatrix} Q & * & * & * & * & * \\ (1+\alpha_1)^{\frac{1}{2}}(A_i + B_i Y) & Q & * & * & * & * \\ S^{\frac{1}{2}}Q & 0 & \gamma I & * & * & * \\ [(1+\alpha_2)R]^{\frac{1}{2}}Y & 0 & 0 & \gamma I & * & * \\ \left[\epsilon\left(1+\alpha_1^{-1}\right)\right]^{\frac{1}{2}}B_j Y & 0 & 0 & 0 & \xi_1 I & * \\ \left[\epsilon\left(1+\alpha_2^{-1}\right)\right]^{\frac{1}{2}}Y & 0 & 0 & 0 & 0 & \xi_2 I \end{bmatrix} \geq 0, j = \{1, 2, \dots, r\} \tag{28}$$

$$Q - \xi_1 I > 0$$
$$Q - \xi_2 I > 0$$

**Proof of Theorem 1.** See Appendix A. □

### 4.2. Nonlinear Systems Subject to Actuator Saturation

In this section, we will extend the preceding development to the case of nonlinear systems under actuator saturation. With Theorem 1 and the Lipschitz condition (Equation (9)), the following theorem is the second result of our paper.

**Theorem 2.** *Consider the discrete-time nonlinear system (Equation (10)) and assume $x(k|k)$ is the measured state of on the infinite horizon quadratic performance $x(k)$ at each sample time $k$. The state feedback control law $u(k + i|k) = H(k)x(k + i|k)$, $i \geq 0$ that minimizes the upper bound $\gamma$ on the infinite horizon quadratic performance index $J(k)$ is given by $H(k) = YQ^{-1}$, where $Q$, $Q > 0$, and $Y$ are the solutions to the following LMIs:*

$$\begin{array}{c} min \ \gamma \\ \gamma, \xi_1, \xi_2, Q, Y \end{array} \tag{29}$$

*Subject to LMI*

$$\begin{bmatrix} I & * \\ x(k) & Q \end{bmatrix} \geq 0 \tag{30}$$

$$
\begin{bmatrix}
Q & * & * & * & * & * & * \\
[(1+\alpha_1)(1+\alpha_2)]^{\frac{1}{2}}(A_i + B_i Y) & Q & * & * & * & * & * \\
S^{\frac{1}{2}}Q & 0 & \gamma I & * & * & * & * \\
[(1+\alpha_3)R]^{\frac{1}{2}}Y & 0 & 0 & \gamma I & * & * & * \\
\left(1+\alpha_1^{-1}\right)^{\frac{1}{2}}N & 0 & 0 & 0 & \xi_1 I & * & * \\
\left[\epsilon(1+\alpha_1)\left(1+\alpha_2^{-1}\right)\right]^{\frac{1}{2}}B_j Y & 0 & 0 & 0 & 0 & \xi_1 I & * \\
\left[\epsilon\left(1+\alpha_3^{-1}\right)\right]^{\frac{1}{2}}Y & 0 & 0 & 0 & 0 & 0 & \xi_2 I
\end{bmatrix} \geq 0, j = \{1, 2, \ldots, r\}
$$

$$
Q - \xi_1 I > 0
$$
$$
Q - \xi_2 I > 0
$$

(31)

Actually, Equation (29) is for constructing an invariant ellipsoid, and Equations (30) and (31) are for guaranteeing robust stability. The symbol "$*$" depicts a symmetric structure.

**Proof of Theorem 2.** See Appendix A. □

## 5. Illustrative Results

### 5.1. Example 1

To demonstrate the effectiveness of the proposed RMPC approach, we consider the discrete-time double integrator [58]:

$$
x(k+1) = \begin{bmatrix} 1 & 1 \\ 1 & 0 \end{bmatrix} x(k) + \begin{bmatrix} 0 \\ 1 \end{bmatrix} u(k), \ y(k) = \begin{bmatrix} 1 & 0 \end{bmatrix} x(k)
$$

which is subject to the input constraint $-1 \leq u(k) \leq 1$. The discrete-time system starts from the initial condition $x(0) = \begin{bmatrix} 10 & -5 \end{bmatrix}^T$. We use our saturated RMPC algorithm (Equation (29)) to cope with this problem and compare the results with the saturated RMPC algorithm presented by Cao and Lin [43] and Kothare et al. [37].

Figure 1 shows the state responses and control input of the system under the three different designs. These responses demonstrate that our RMPC guarantees better stabilization performance than that of the previous designs, where the proposed controller results in a smaller overshoot, a shorter rising time, and a faster time response.

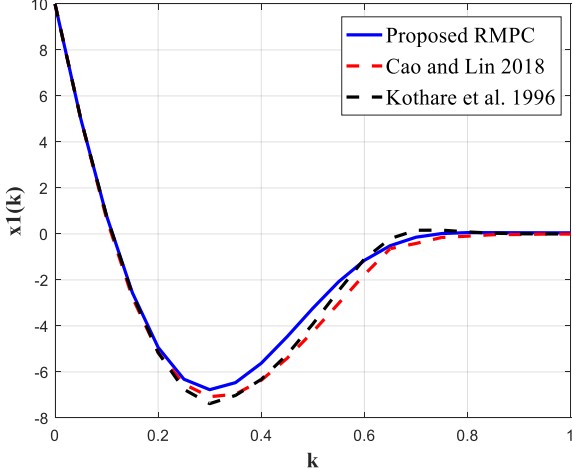

**Figure 1.** *Cont.*

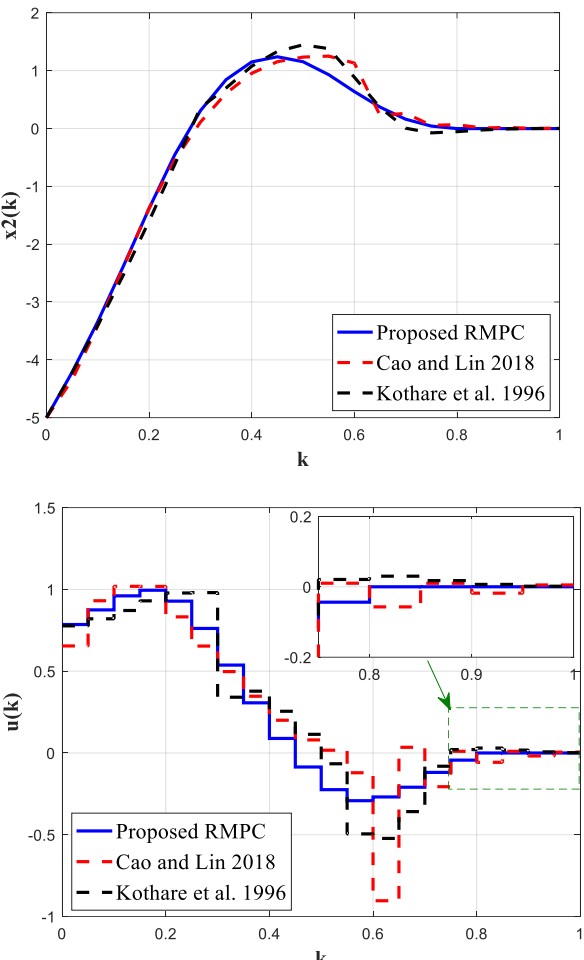

**Figure 1.** State responses and control input [37,43].

Moreover, Figure 2 shows that the accumulated costs associated with the proposed RMPC algorithm is lower than those of the other designs.

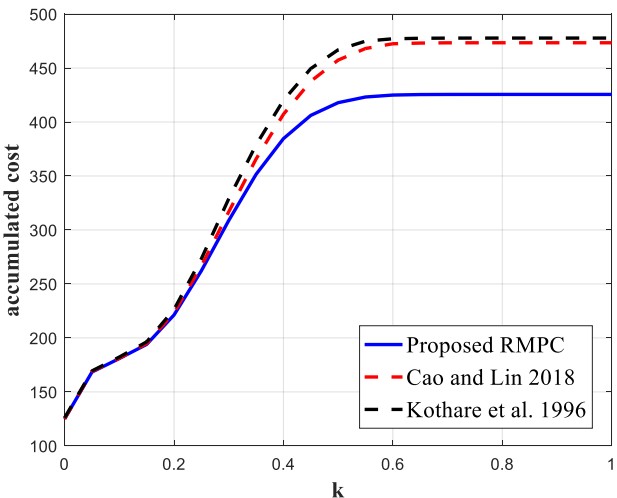

**Figure 2.** Accumulated costs [37,43].

*5.2. Example 2. DC–DC Buck-Converter-Based PV Pumping System*

The photovoltaic pumping system under consideration is shown in Figure 3. It comprises a PVG, a DC–DC buck converter, and a DC motor coupled to a centrifugal pump.

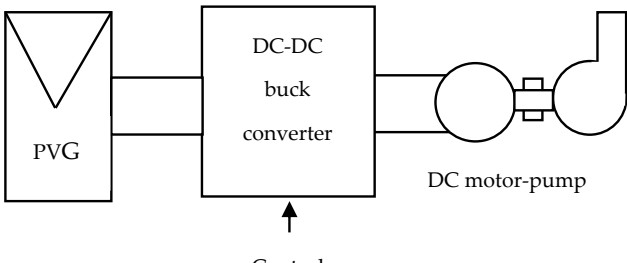

**Figure 3.** Proposed PV pumping system block diagram.

5.2.1. PVG Model

Generally, a PVG consists of several solar cells, which are connected in series and parallel to achieve the required voltage and current. The single-diode circuit model in Figure 4a is the most widely used one to study the behavior of a PVG, in which the PVG can be modeled as a current source to model the photo-current $I_{ph}$ in parallel with a diode $D$, an intrinsic shunt resistance $R_P$, and a series resistance $R_S$ (more details can be found in [59,60]). A PVG has a nonlinear voltage–current $(V - I)$ characteristic (see Figure 5) given by Equation (32), where $v_{pv}$ and $I_{PV}$ are the output voltage and current of the PV module, respectively. The efficiency of PVG depends on the internal characteristics of the device $R_P$ and $R_S$ and on many ambient conditions, such as the solar irradiance level, temperature, and shaded condition.

$$I_{PV} = I_{ph} - I_0 \left[ exp\left( \frac{v_{pv} + I_{PV}R_S}{nV_t} \right) - 1 \right] - \left( \frac{v_{pv} + I_{PV}R_S}{R_P} \right) \tag{32}$$

$I_0$ is the reverse saturation current.
$V_t = \frac{N_s KT}{q}$ is the junction thermal voltage of the array with $N_s$ cells connected in series.
$n$ is the ideality factor of the PV cell.
$q = 1.60217646 \times 10^{-19}$ C is the electron charge.
$K = 1.3806503 \times 10^{-23}$ J/K is the Boltzmann constant.
T is the temperature in degrees Kelvin.
Since $R_p \gg R_S$, it is possible to assume that $I_{sc} \approx I_{mpp}$. Thus, the photo-current for any temperature and solar irradiation can be expressed by the following equation [61]:

$$I_{ph} = \frac{G_0}{G_{ref}} \left[ I_{mpp} + K_i \left( T - T_{ref} \right) \right] \tag{33}$$

$I_{sc}$ is the short-circuit current.
$I_{mpp}$ is the MPP current at STC.
$K_i$ is the short-circuit current temperature coefficient.
$G_0$ is the solar irradiance in W/m$^2$.
$G_{ref} = 1000$ W/m$^2$ is the solar irradiation reference at STC.
$T_{ref} = 25$ °C (298 degrees Kelvin) is the PVG temperature reference at STC.
The cell reverse saturation current, which depends on temperature, is given by:

$$I_0 = I_{rs} \left[ \frac{T}{T_{ref}} \right]^3 exp\left[ \frac{qE_g}{nK} \right] \tag{34}$$

$I_{rs}$ is the reverse saturation current.
$E_g = 1.12$ eV is the band-gap energy of the semiconductor used in the cell.

$$I_{rs} = \frac{I_{sc,n}}{exp\left( \frac{V_{oc,n}}{nV_{t,n}} \right) - 1} \tag{35}$$

$I_{sc,n}$ is the cell short-circuit current at STC.

$V_{oc,n}$ is the open-circuit voltage at STC.

$V_{t,n}$ is the junction thermal voltage with $N_s$ cells connected in series at $T_{ref}$.

A modification on the reverse saturation current is proposed in [61] to match the open-circuit voltages of the model with the experimental data (from the manufacturer datasheet) for a wide range of temperatures

$$I_0 = \frac{I_{sc,n} + K_i\left(T - T_{ref}\right)}{exp\left[\frac{V_{oc,n} + K_v\left(T - T_{ref}\right)}{nV_t}\right] - 1} \tag{36}$$

$K_v$ is the open-circuit voltage temperature coefficient.

The relation between $R_p$ and $R_S$ can be determined by solving Equation (33) for $R_S$ [61]:

$$P_{max} = I_{mpp}V_{mpp} = V_{mpp}\left[I_{ph} - I_0\left[exp\left(\frac{q}{KT}\frac{V_{mpp} + I_{mpp}R_S}{N_s n}\right) - 1\right] - \frac{V_{mpp} + I_{mpp}R_S}{R_p}\right] \tag{37}$$

$P_{max}$ is the maximum power from the manufacturer datasheet. Then, $R_p$ can be derived iteratively as:

$$R_p = \frac{V_{mpp}\left(V_{mpp} + I_{mpp}R_S\right)}{V_{mpp}I_{ph} - V_{mpp}I_0 exp\left[\frac{q\left(V_{mpp} + I_{mpp}R_S\right)}{N_s nKT}\right] + V_{mpp}I_0 - P_{max}} \tag{38}$$

To simplify the PVG model, a Thévenin equivalent circuit shown in Figure 4b can be used, with:

$$R_{TH} = R_S + R_P \parallel R_D \tag{39}$$

$$V_{TH} = I_{PV}R_P \tag{40}$$

where $V_{TH}$ is Thévenin equivalent voltage and $R_{TH}$ is Thévenin resistance. It can be observed that the Thévenin equivalent circuit parameters are not constant but depend on the environmental variables and operating point.

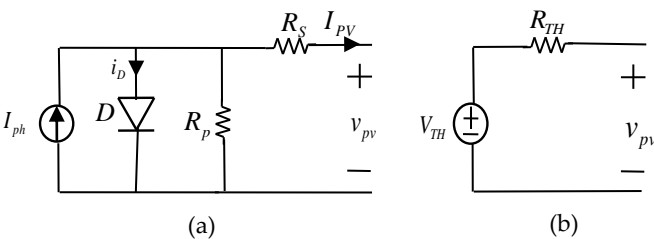

(a)                     (b)

**Figure 4.** Equivalent electrical scheme of the PVG: (**a**) detailed and (**b**) Thévenin.

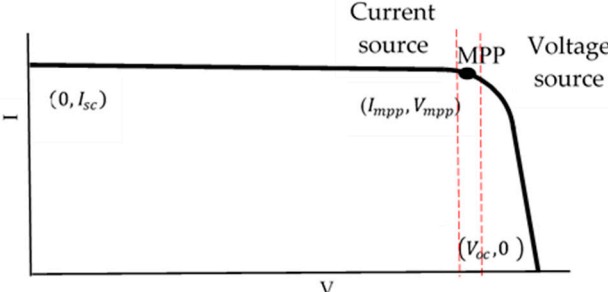

**Figure 5.** PVG I–V curve.

### 5.2.2. DC Motor-Pump Model

The DC motor in Figure 3 is a DC motor with a permanent magnet (see Figure 6). This model is defined by an electrical circuit equation armature:

$$v_a(t) = R_a i_a(t) + e(t) + L_a \frac{di_a(t)}{dt} \tag{41}$$

where $e(t)$ is the counter electromotive force, $i_a$ is the armature current, $v_a$ is the armature voltage, and $R_a$ and $L_a$ represent the armature resistance and inductance, respectively.

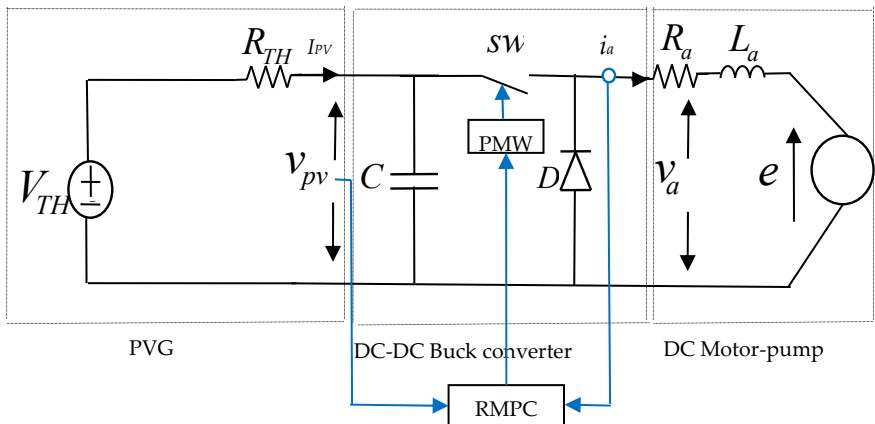

PVG          DC-DC Buck converter          DC Motor-pump

**Figure 6.** Equivalent electrical scheme of the PV pumping system with the RMPC.

### 5.2.3. Buck Converter Model

Figure 5 shows the schematic circuit diagram of a PV pumping system formed after connecting a DC motor to a DC–DC buck converter fed from a photovoltaic generator. The converter is assumed to operate in continuous conduction mode (CCM), and the inductor current is always larger than 0. Control is implemented via a pulse width modulation (PWM) approach. $v_{pv}$ is the photovoltaic array voltage, which must be controlled through variation of the duty cycle $d(t)$ in order to keep the array operation at the maximum power point. The diode $D$ is on inverse polarization, while $C$ represents the capacitor value, and $sw$ is a power MOSFET controlled by a binary signal $S_b(t)$; see Figure 7. The binary signal that triggers the switches on and off is controlled by a fixed-frequency pulse width modulation (PWM) circuit (Figure 7). $1/T_s$ is the constant switching frequency of the PWM circuit, and $T_s$ is the switching period given by:

$$T_s = T_{on} + T_{off} \tag{42}$$

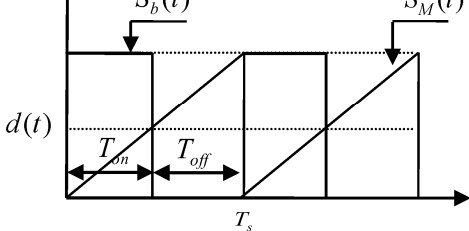

**Figure 7.** PWM waveforms.

$T_{on}$ is the time when the MOSFET is on ($u_b = 1$). $T_{off}$ is the time when the MOSFET is off ($u_b = 0$). The ratio $T_{on}/(T_{on} + T_{off})$ is the duty cycle $d(t)$. A PWM signal of constant frequency can be obtained by comparing the duty cycle with a sawtooth

signal $S_M(t)$ of amplitude equal to 1. Consequently, the duty cycle is constrained in amplitude between 0 and 1:

$$0 \leq d(t) \leq 1 \tag{43}$$

The average circuit model of the PV pumping system is shown in Figure 8. The corresponding discrete-time model is obtained by using the forward Euler approximation as follows:

$$
\begin{aligned}
i_a(k+1) &= \left(1 - \frac{T_S R_a}{L_a}\right) i_a(k) - \frac{T_S}{L_a} e(k) + \frac{T_S}{L_a} v_{PV}(k) d(k) \\
v_{PV}(k+1) &= -\frac{T_S}{C} i_a(k) d(k) + \left(1 - \frac{T_S}{CR_{TH}}\right) v_{PV}(k) + \frac{T_S V_{TH}}{R_{TH}}
\end{aligned}
\tag{44}
$$

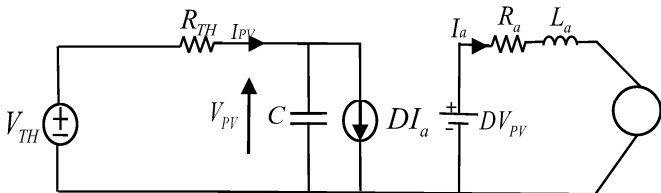

**Figure 8.** Averaged circuit model of PV pumping system.

The total instantaneous quantities of the PV system are presented as the sum of the DC and AC components.

$$
\begin{aligned}
x(k) &= \tilde{x}(k) + X \\
d(k) &= \tilde{d}(k) + D
\end{aligned}
\tag{45}
$$

where $x = \left[i_a(k), \ v_{pv}(k)\right]^T \in \mathbb{R}^2$ is the state, $d(\mathrm{k}) \in \mathbb{R}$ is the saturated control input of the buck converter, $X$ and $D$ represent the equilibrium values, and $\tilde{x}$ and $\tilde{d}$ are the perturbed values of state and input. Using the same concept, we can obtain:

$$
\begin{aligned}
i_a(k) &= \tilde{i}_a(k) + I_a \\
v_{PV}(k) &= \tilde{v}_{PV}(k) + V_{PV} \\
d(k) &= H\tilde{x}(k) + HX
\end{aligned}
\tag{46}
$$

Equation (46) is substituted into Equation (44), and a small-signal model is derived as follows:

$$
\begin{aligned}
\tilde{i}_a(k+1) &= \left(1 - \frac{T_S R_a}{L_a}\right) \tilde{i}_a(k) - \frac{T_S V_{PV}}{L_a} \tilde{d}(k) + \frac{T_S D}{L_a} \tilde{v}_{PV}(k) \\
\tilde{v}_{PV}(k+1) &= -\frac{T_S I_a}{C} \tilde{d}(k) - \frac{T_S D}{C} \tilde{i}_a(k) + \left(1 - \frac{T_S}{CR_{TH}}\right) \tilde{v}_{PV}(k)
\end{aligned}
\tag{47}
$$

Equation (47) can be expressed as follows:

$$x(k+1) = \tilde{x}(k) + \dot{X} = A\tilde{x}(k) + AX + B_{(0,1)}^{sat}(H\tilde{x}(k) + HX) + \tilde{f}\left(\tilde{x}(k), \tilde{d}(k)\right) \tag{48}$$

where $\tilde{f}$ is the nonlinear term that is obtained by differing between the nominal nonlinear model and the linear part, given by:

$$\tilde{f}\left(\tilde{x}(k), \tilde{d}(k)\right) = f(x(k), d(k)) - A\tilde{x}(k) + AX + B_{(0,1)}^{sat}(H\tilde{x}(k) + HX) \tag{49}$$

And $A = \begin{bmatrix} 1 - \frac{T_S R_a}{L_a} & \frac{T_S D}{L_a} \\ -\frac{T_S D}{C} & 1 - \frac{T_S}{CR_{TH}} \end{bmatrix}$, $B = \begin{bmatrix} \frac{T_S V_{PV}}{L_a} \\ -\frac{T_S I_a}{C} \end{bmatrix}$.

The control input is subject to non-symmetric actuator saturation, and we can use the method proposed in [25] to transform it to a symmetric saturation. Thus, the saturation function of the control input is rewritten as follows:

$$\underset{(0,1)}{sat}\left(H\widetilde{x}(k) + HX\right) = \begin{cases} 0, & if & H\widetilde{x}(k) + HX < 0 \\ H\widetilde{x}(k) + HX & if & 0 \leq H\widetilde{x}(k) + HX \leq 1 \\ 1, & if & H\widetilde{x}(k) + HX > 1 \end{cases} \tag{50}$$

which can be expressed as

$$\underset{(-HX,1-HX)}{sat}(K\widetilde{x}(k)) = \begin{cases} HX, & if & H\widetilde{x}(k) < -HX \\ H\widetilde{x}(k) & if & HX \leq H\widetilde{x}(k) \leq 1 - HX \\ 1 - HX, & if & H\widetilde{x}(k) + HX > 1 - HX \end{cases} \tag{51}$$

If we add $HX$ to Equation (48), we can obtain the following equality:

$$\underset{(0,1)}{sat}\left(H\widetilde{x}(k) + HX\right) = \underset{(-HX,1-HX)}{sat}\left(H\widetilde{x}(k)\right) + HX \tag{52}$$

Since the steady-state part is equal to 0, there is:

$$f(X,D) = AX + HX = 0 \tag{53}$$

In our case, the steady-state control signal is $HX = 0.5$, and the operating point can be calculated from:

$$R_a I_a = V_{PV} D$$
$$D I_a = (-V_{PV} + V_{TH})/R_{TH} \tag{54}$$

The system described in Equation (48) can be written as follows:

$$\widetilde{x}(k+1) = A\widetilde{x}(k) + Bsat\left(\widetilde{\delta}(k)\right) + \widetilde{f}\left(\widetilde{x}(k), \widetilde{d}(k)\right) \tag{55}$$

where $sat\left(\widetilde{\delta}(k)\right) = \underset{(-HX,1-HX)}{sat}\left(H\widetilde{x}(k)\right)$ is the new control input. Lemma 1 can be used to handle saturation nonlinearity. Thus, the new symmetric saturation model is constrained as in [62]:

$$-U_0 < u(k) < U_0 \leftrightarrow -HX < \delta(k) < HX \tag{56}$$

Based on this amplitude saturation, the decentralized dead-zone nonlinearity can be defined as:

$$\widetilde{\varphi}(k) = sat\left(\widetilde{\delta}(k)\right) - \widetilde{\delta}(k) \tag{57}$$

The closed-loop system is rewritten as:

$$\widetilde{x}(k+1) = (A + BH)\widetilde{x}(k) + B\widetilde{\varphi}(k) + \widetilde{f}\left(\widetilde{x}(k), \widetilde{d}(k)\right) \tag{58}$$

### 5.2.4. Uncertainty Polytope Model

In this example, the Thévenin resistance $R_{TH}$ at the operating point is considered as the uncertain parameter. The discrete state-space model described in Equation (58) is expressed as:

$$\widetilde{x}(k+1) = (A(\beta) + BH)\widetilde{x}(k) + B\widetilde{\varphi}(k) + \widetilde{f}\left(\widetilde{x}(k), \widetilde{d}(k)\right) \tag{59}$$

We consider $r = 2$ and the vector $\beta[1/R_{TH}]$, in which:

$$\frac{1}{R_{TH}} \in \left[\frac{1}{R_{THmin}}, \frac{1}{R_{THmax}}\right] \tag{60}$$

Photovoltaic generators are neither constant voltage sources nor constant current sources. In practice, the PVG is forced to operate at the boundaries of the constant current and constant voltage modes, if an MPPT is used, and the possessing dynamic resistance is equal to the resistance of its load at the maximum power transfer [63]. Based on the PVG equivalent circuit of Figure 4b, we can define a range of changes for Thévenin resistance. Under the open-circuit condition, $R_D$ is low, dominating the parallel connection with $R_P$ [15]. Therefore, we have:

$$R_{TH}|_{oc} \rightarrow R_S + R_D \tag{61}$$

bounded by:

$$R_{TH}|_{oc} > R_{THmin} = R_S \tag{62}$$

With short-circuit and reference conditions, $R_D$ is high and $R_P$ dominates the parallel connection. There is:

$$R_{TH}|_{sc} \rightarrow R_P + R_S \tag{63}$$

Note that $R_S$ is constant and $R_P$ is irradiation dependent [64]:

$$\frac{R_P}{R_{P,ref}} = \frac{G_0}{G_{ref}} \tag{64}$$

where $G_0$ is the solar irradiance and $R_{P,ref}$ is the shunt resistance at STC (the solar irradiation is $G_{ref} = 1000 \text{ W/m}^2$, and the PVG temperature is $T_{ref} = 25 \,°C$). As an approximation, we have:

$$R_{TH}|_{sc} < R_{THmax} = R_{P,STC} \tag{65}$$

Since the PV system matrix $A(\beta)$ depends linearly on the uncertain parameter $1/R_{TH}$, we can define a polytope of $r = 2$ vertices, which contains all the possible values of the uncertain matrix. The uncertain matrices $A_j$ are:

$$A_1 = \begin{bmatrix} 1 - \frac{T_S R_a}{L_a} & \frac{T_S D}{L_a} \\ -\frac{T_S D}{C} & 1 - \frac{T_S}{C R_p} \end{bmatrix}, \ A_2 = \begin{bmatrix} 1 - \frac{T_S R_a}{L_a} & \frac{T_S D}{L_a} \\ -\frac{T_S D}{C} & 1 - \frac{T_S}{C R_S} \end{bmatrix}$$

### 5.2.5. Simulations Results

In this section, several simulation tests are implemented to verify the performance of the proposed control law and its robustness according to the presence of the input constraint, insolation variations, temperature variations, and dynamic resistance uncertainty. The results are compared to the RMPC method for nonlinear systems dependent on the Lipschitz bound presented in Poursafar et al. [55] and the saturated RMPC algorithm presented in Cao and Lin [43] designed by the linearized model around the equilibrium point. The parameters of the DC–DC converter and the PVG can be found in Tables 1 and 2, respectively.

**Table 1.** Buck converter parameters.

| Parameters | Description | Numerical Value |
|---|---|---|
| L | Inductance | 1 mH |
| C | Input capacitance | 1000 µF |
| $V_a$ | Armature voltage | 13.5 V |
| Ia | Armature current | 15.22 A |
| D | Duty cycle | 0.5 |
| $T_s$ | Switching period | 0.654 ms |

**Table 2.** PVG parameters at STC.

| Parameters | Description | Numerical Value |
|---|---|---|
| $P_{max}$ | Maximum power | 200.143 W |
| $V_{mpp}$ | Voltage at $P_{max}$ | 26.3 V |
| $I_{mpp}$ | Current at $P_{max}$ | 7.61 A |
| $I_{sc}$ | Short-circuit current | 8.21 A |
| $V_{oc}$ | Open-circuit voltage | 32.9 V |
| $K_V$ | Open-circuit voltage temperature coefficient | $-0.1230$ V/K |
| $K_i$ | Short-circuit current temperature coefficient | 0.0032 A/K |
| $R_{TH}$ | Thévenin resistance | 415.405 $\Omega$ |
| $R_s$ | Series resistance | 10 $\Omega$ |
| $R_{p,STC}$ | Shunt resistance at STC | 405.405 $\Omega$ |

- *Scenario 1: The input saturation effect* In the first case, we assume that the saturation limit is $\delta_{max} = 0.5$, and the initial conditions of the DC–DC buck converter during startup are represented by $x_0 = [15.22, 26.3]^T$. We use the nominal value of the dynamic resistance $R_{TH} = 415.405\ \Omega$ to test the closed-loop behavior without any change in the converter parameters. Figure 9 depicts the computed input and state responses under the three different designs, where the waveforms are the PV voltage $v_{pv}$, armature current $i_a$, and duty cycle $d$. From Figure 9, it can be discovered that the proposed method outperforms the saturated RMPC algorithm in Cao and Lin [43] with a fairly good time response and lower fluctuations. Moreover, the PV voltage response settles to its desired value without any overshoot, whereas the approach in [55] has some unstable transient responses. In other words, our RMPC is well capable of handling the hard actuator saturation constraint.

- *Scenario 2: The insolation variations effect* In the second case, we assume that the saturation limit is $\delta_{max} = 0.5$. An examination is made for various irradiances, such as $G_0 = 200\ \text{W/m}^2$, $G_0 = 400\ \text{W/m}^2$, and $G_0 = 1000\ \text{W/m}^2$. With the proposed control method, the power regulation response is shown in Figure 10. The increasing and decreasing nature of PV power with respect to insolation $G_0$ can be observed in the waveforms of PV power, which verifies the PV generator voltage at the MPP tracking. Similarly, the proposed RMPC also shows a good time response, low oscillation, and desired stability.

- *Scenario 3: The temperature variations effect* In this scenario, we consider varying temperature with constant insolation $G_0 = 1000\ \text{W/m}^2$. The temperature changes between 100 °K and 298 °K at $t = 100$ s. Next, the temperature increases to 400 °K at $t = 200$ s, and it returns to 100 °K at $t = 300$ s, For this scenario, the saturation limit is $\delta_{max} = 0.5$. Figure 11 depicts the computed state responses under the proposed RMPC design. It is easily seen that the proposed RMPC guarantees the desired stability, with a good time response and low oscillation.

- *Scenario 4: The dynamic resistance variations effect* In the last case, we consider dynamic resistance uncertainty in order to validate the robustness of the proposed controller. Figure 12 illustrates that our controller can stabilize the system on the reference PV voltage in presence of an abrupt dynamic resistance change. The overshoots and long settling time at $t = 50$ s and $t = 120$ s are the results of the aggressive move in the converter parameters, which demonstrates the effectiveness of our MPC design algorithm.

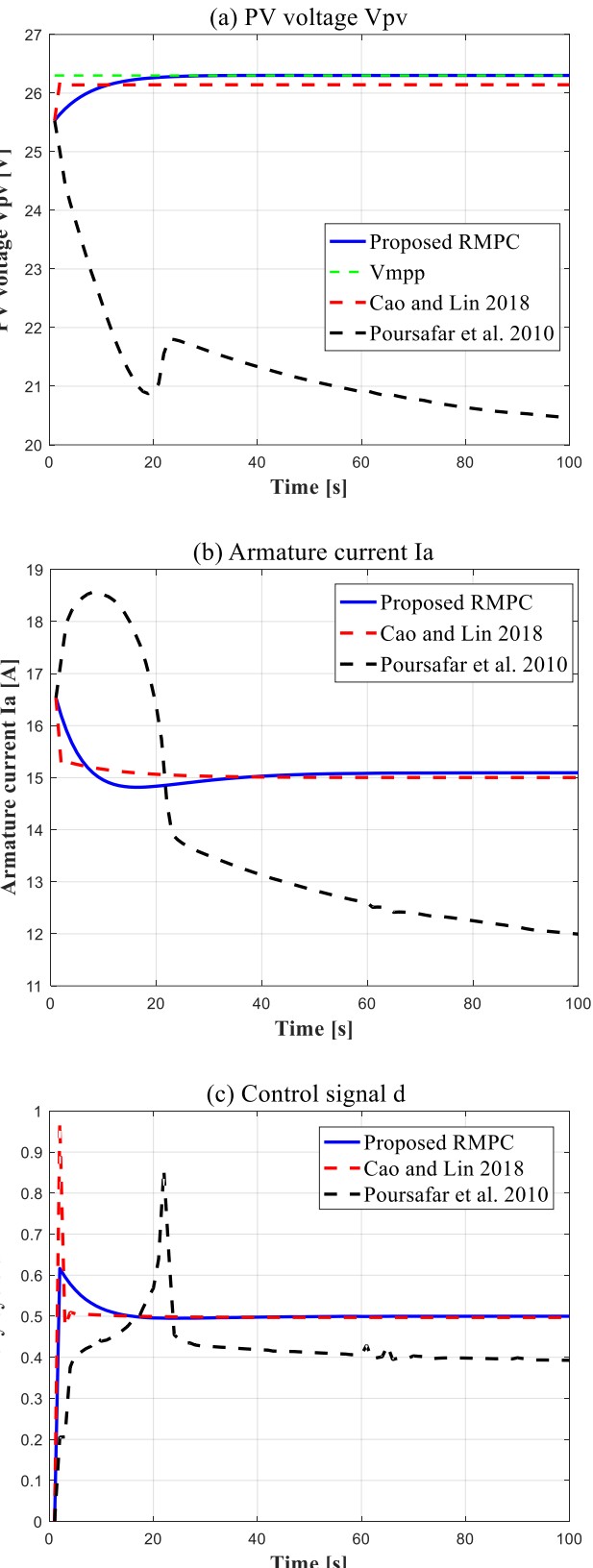

**Figure 9.** Simulated transient of the buck converter under saturation constraint $|\delta_{max}| \leq 0.5$: (**a**) PV voltage $V_{pv}$, (**b**) armature current $I_a$, and (**c**) control signal $d$.

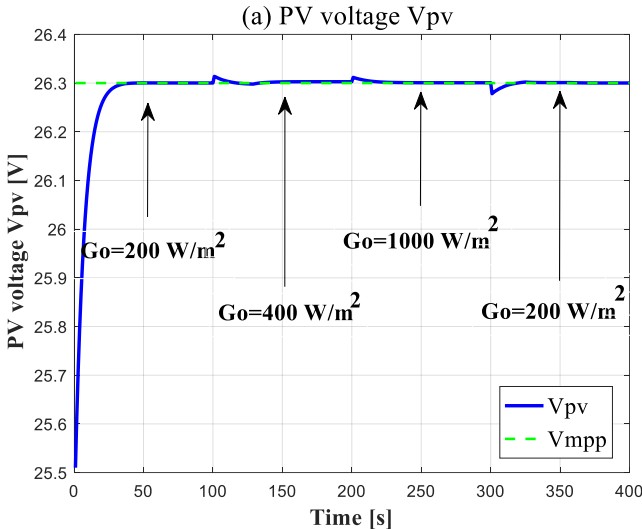

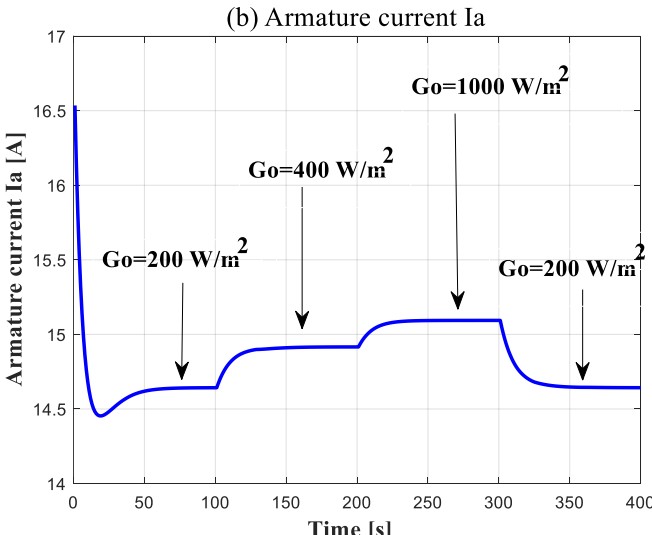

**Figure 10.** Simulated transient of the buck converter with varying insolation: (**a**) PV voltage $V_{pv}$ and (**b**) armature current $I_a$.

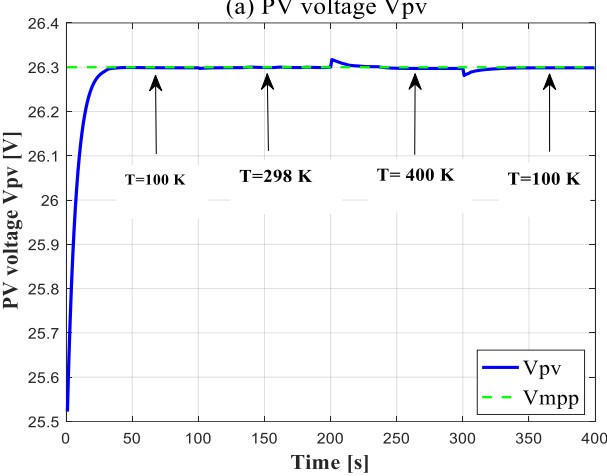

**Figure 11.** *Cont.*

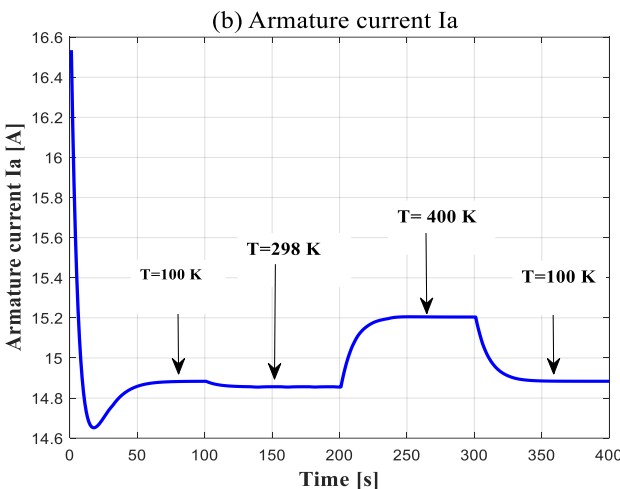

**Figure 11.** Simulated transient of the buck converter with varying temperature: (**a**) PV voltage $V_{pv}$ and (**b**) armature current $I_a$.

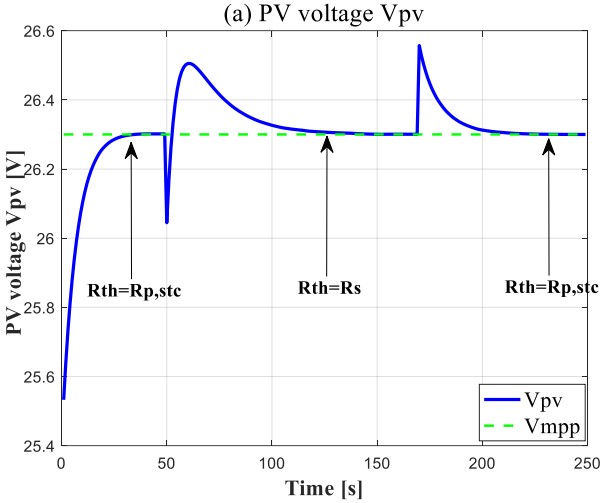

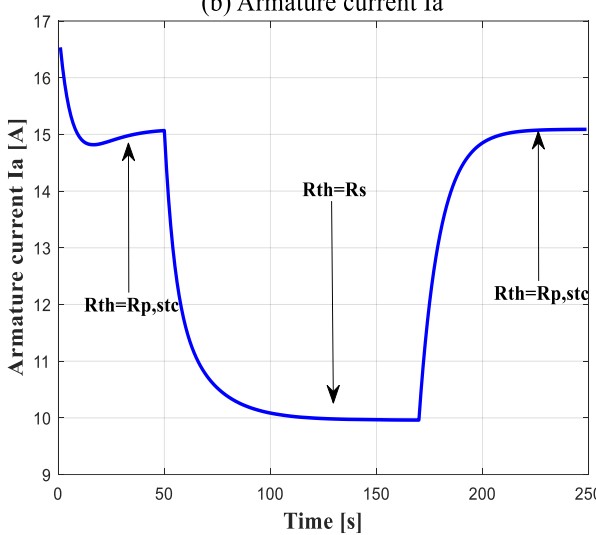

**Figure 12.** Simulated transient of the buck converter under a dynamic resistance step transient $[R_{TH} = R_{P,STC} \rightarrow R_S \rightarrow R_{P,STC}]$: (**a**) PV voltage $V_{pv}$ and (**b**) armature current $I_a$.

## 6. Conclusions

This paper proposes a new RMPC design framework based on Lyapunov stability theory for a PV pumping system with actuator saturation and uncertain parameters. The input saturation effect is expressed by a model using dead-zone nonlinearity. At each time instant, the state feedback control law is obtained by minimizing the upper bound of the infinite horizon cost function within the framework of LMIs, which directly incorporates the input saturation. The internal stability of the closed-loop system is guaranteed in the sense of Lyapunov stability. The design of the closed-loop stabilization for a nonlinear system for the requirement of desired output responses with uncertain parameters is also investigated here. The simulation results obtained demonstrate the effectiveness and efficiency of our method. Furthermore, the novel method is examined in case the uncertainty is in the form of solar irradiance change and dynamic resistance, which proves that it is able to stabilize the system at the desired PV voltage.

**Author Contributions:** Conceptualization, O.H., S.B., F.A. and M.C. (Mohammed Chadli); methodology, O.H., S.B., M.C. (Mohammed Chadli) and M.C. (Mohamed Chemachema); validation, O.H., S.B., F.A., M.C. (Mohammed Chadli) and M.C. (Mohamed Chemachema); formal analysis, O.H., S.B., I.B. and M.C. (Mohammed Chadli); investigation, O.H., S.B., I.B. and M.C. (Mohammed Chadli); writing—original draft preparation, S.B., I.B., B.N. and M.C. (Mohamed Chemachema); writing—review and editing, S.B., I.B., B.N. and M.C. (Mohammed Chadli). All authors have read and agreed to the published version of the manuscript.

**Funding:** This research received no external funding.

**Institutional Review Board Statement:** Not applicable.

**Informed Consent Statement:** Not applicable.

**Data Availability Statement:** Not applicable.

**Conflicts of Interest:** The authors declare no conflict of interest.

## Nomenclature

**Abbreviations**

| | |
|---|---|
| MPC | model predictive controller |
| RMPC | robust model predictive controller |
| DC | direct current |
| DC–DC | direct current–direct current |
| PV | photovoltaic |
| PVG | photovoltaic generator |
| LMI | linear matrix inequality |
| $H_2O_2$ | hydrogen peroxide |
| $CO_2$ | carbon dioxide |
| MPP | maximum power point |
| MPPT | maximum power point tracker |
| P&O | perturb and observe |
| INC | incremental conductance |
| STC | standard test condition |
| PID | proportional–integral–derivative |
| SMC | sliding mode control |
| LPV | linear parameter varying |
| CNC | computer numerical control |
| AMB | active magnetic bearing |
| CCM | continuous conduction mode |
| PWM | pulse width modulation |

**Symbols**

| | |
|---|---|
| $x \in \mathbb{R}^n$ | state |
| $u \in \mathbb{R}^m$ | control input |
| $\overline{u} \in \mathbb{R}^m$ | saturation function of control input |
| $k$ | current time instant |
| $u_{lim}$ | control input limit |
| $u_{max}$ | maximum control input limit |
| $f \in \mathbb{C}^2$ | nonlinear function of states and control inputs |
| $\beta_j$ | uncertain parameter |
| $\Omega$ | convex hull |
| $\widetilde{f}(x,\overline{u})$ | globally Lipschitz |
| $N \in \mathbb{R}^{n \times n}$ | Lipschitz constant matrix |
| $I$ | identity matrix |
| $H$ | state feedback gain |
| $\varphi_i$ | dead-zone nonlinearity function |
| $J$ | worst-case performance function |
| $S \in \mathbb{R}^{n \times n}$ | positive definite state weight |
| $R \in \mathbb{R}^{m \times m}$ | positive definite control weight |
| $V(x)$ | quadratic Lyapunov function |
| $\gamma$ | positive scalar |
| $I_{PV}$ | output current of PV generator, in ampere |
| $v_{pv}$ | output voltage of PV generator, in volt |
| $I_{ph}$ | photo-current of the PV generator |
| $I_0$ | reverse saturation current |
| $I_{sc}$ | short-circuit current |
| $K_i$ | short-circuit current temperature coefficient |
| $E_g$ | band $-$ gap energy of the semiconductor used in the cell, in eV |
| $I_{sc,n}$ | cell short-circuit current at STC |
| $V_{oc,n}$ | open-circuit voltage at STC |
| $V_{t,n}$ | junction thermal voltage at $T_{ref}$ |
| $K_v$ | open-circuit voltage temperature coefficient |
| $I_{rs}$ | reverse saturation current |
| $R_P$ | shunt resistance, in ohm |
| $R_S$ | series resistance, in ohm |
| $V_t$ | junction thermal voltage |
| $n$ | ideality factor of the PV cell |
| $q$ | electron charge, in coulomb |
| $K$ | Boltzmann contant, in J/K |
| $i_D$ | recombination losses current |
| $V_{TH}$ | Thévenin equivalent voltage, in volt |
| $R_{TH}$ | Thévenin resistance, in ohm |
| $e(t)$ | counter electromotive force |
| $i_a$ | DC motor armature current, in ampere |
| $v_a$ | DC motor armature voltage, in volt |
| $R_a$ | armature resistance, in ohm |
| $L_a$ | armature inductance, in henry |
| $D$ | diode |
| $d$ | converter duty cycle |
| $C$ | capacitance, in farad |
| $S_b$ | binary signal |
| $T_s$ | witching period, in second |
| $S_M$ | sawtooth signal |
| $\widetilde{x}$ | perturbed values of state |
| $\widetilde{d}$ | perturbed values of input |
| $V_{MPP}$ | voltage at MPP, in volt |
| $I_{MPP}$ | current at MPP, in ampere |

| | |
|---|---|
| $P_{max}$ | MPP from the manufacturer datasheet, in watt |
| $R_{P,ref}$ | shunt resistance at SRC, in ohm |
| $G_0$ | solar irradiance, in W/m$^2$ |
| $G_{ref}$ | solar irradiation reference, in W/m$^2$ |
| T | PVG temperature, in degrees Kelvin |
| $T_{ref}$ | PVG temperature reference, in degrees Kelvin |

## Appendix A  Proof of Theorem 2

To obtain the LMI (Equation (27)), $V(x)$ is required to satisfy:

$$V(k+i+1|k) - V(k+i|k) \leq -(x(k+i|k)^T S x(k+i|k) + \overline{u}(k+i|k)^T R \overline{u}(k+i|k)) \quad \text{(A1)}$$

By substituting Equation (10) in Equation (A1), we have:

$$\left[ (A_j + B_j H)x(k+i|k) + B_j\varphi(k+i|k) + \widetilde{f}(x(k+i|k), \overline{u}(k+i|k)) \right]^T$$
$$\times P\left[ (A_j + B_j H)x(k+i|k) + B\varphi(k+i|k) + \widetilde{f}(x(k+i|k), \overline{u}(k+i|k)) \right]$$
$$- x(k+i|k)^T P x(k+i|k) + x(k+i|k)^T S x(k+i|k) + \overline{u}(k+i|k)^T R \overline{u}(k+i|k) \leq 0$$

Define the function $h_1(x, \overline{u})$ as:

$$h_1(x, \overline{u}) = \left[ (A_j + B_j H)x(k+i|k) + B_j\varphi(k+i|k) + \widetilde{f}(x(k+i|k), \overline{u}(k+i|k)) \right]^T$$
$$\times P\left[ (A_j + B_j H)x(k+i|k) + B_j\varphi(k+i|k) + \widetilde{f}(x(k+i|k), \overline{u}(k+i|k)) \right]$$
$$= \left[ (A_j + B_j H)x(k+i|k) + B_j\psi(k+i|k) \right]^T P\left[ (A_j + B_j H)x(k+i|k) + B_j\varphi(k+i|k) \right]$$
$$+ \left[ (A_j + B_j H)x(k+i|k) + B_j\varphi(k+i|k) \right]^T P[\widetilde{f}(x(k+i|k), \overline{u}(k+i|k))]$$
$$+ [\widetilde{f}(x(k+i|k), \overline{u}(k+i|k))]^T P\left[ (A_j + B_j H)x(k+i|k) + B_j\varphi(k+i|k) \right]$$
$$+ [\widetilde{f}(x(k+i|k), \overline{u}(k+i|k))]^T P[\widetilde{f}(x(k+i|k), \overline{u}(k+i|k))]$$

where applying Lemma 3 on $h_1(x, \overline{u})$ yields:

$$h_1(x, \overline{u}) \leq (1 + \alpha_1)\left[ (A_j + B_j H)x(k+i|k) + B_j\varphi(k+i|k) \right]^T P\left[ (A_j + B_j H)x(k+i|k) \right]$$
$$+ B_j\varphi(k+i|k) + \left(1 + \alpha_1^{-1}\right)[\widetilde{f}(x(k+i|k), \overline{u}(k+i|k))]^T P[\widetilde{f}(x(k+i|k), \overline{u}(k+i|k))]$$

Consider $P \leq \lambda_{1,max} I \leq \mu_1 I$, where $\lambda_{1,max}$ is the maximum eigenvalue of $P$ and $\mu_1 I$ is the corresponding upper bound. There is:

$$h_1(x, \overline{u}) \leq (1 + \alpha_1)\left[ (A_j + B_j H)x(k+i|k) + B_j\varphi(k+i|k) \right]^T$$
$$\times P\left[ (A_j + B_j H)x(k+i|k) + B_j\varphi(k+i|k) \right] \quad \text{(A2)}$$
$$+ \left(1 + \alpha_1^{-1}\right)\mu_1[\widetilde{f}(x(k+i|k), \overline{u}(k+i|k))]^T[\widetilde{f}(x(k+i|k), \overline{u}(k+i|k))]$$

Using Lipschitz property in Equation (9) helps in further simplifying Equation (A2) to:

$$[\widetilde{f}(x(k+i|k), \overline{u}(k+i|k))]^T[\widetilde{f}(x(k+i|k), \overline{u}(k+i|k))] \leq x(k+i|k)^T N^T N x(k+i|k)$$

There is:

$$h_1(x, \overline{u}) \leq (1 + \alpha_1)\left[ (A_j + B_j H)x(k+i|k) + B_j\varphi(k+i|k) \right]^T$$
$$\times P\left[ (A_j + B_j H)x(k+i|k) + B_j\varphi(k+i|k) \right] + \left(1 + \alpha_1^{-1}\right)\mu_1 x(k+i|k)^T N^T N x(k+i|k)$$

Inequality in Equation (A1) becomes:

$$\left(1 + \alpha_1^{-1}\right)\mu_1 x(k+i|k)^T N^T N x(k+i|k) - x(k+i|k)^T P x(k+i|k)$$
$$+ (1 + \alpha_1)\left[(A_j + B_j H)x(k+i|k) + B_j \varphi(k+i|k)\right]^T P\left[(A_j + B_j H)x(k+i|k) \right.$$
$$\left. + B_j \varphi(k+i|k)\right] + x(k+i|k)^T S x(k+i|k) + \overline{u}(k+i|k)^T R \overline{u}(k+i|k) \le 0$$
(A3)

Now, we define another function $h_2(x, \overline{u})$ as:

$$h_2(x, \overline{u}) = \left[(A_j + B_j H)x(k+i|k) + B_j \varphi(k+i|k)\right]^T$$
$$\times P\left[(A_j + B_j H)x(k+i|k) + B_j \varphi(k+i|k)\right]$$
$$= \left[(A_j + B_j H)x(k+i|k)\right]^T P\left[(A_j + B_j H)x(k+i|k)\right] + \left[(A_j + B_j H)x(k+i|k)\right]^T$$
$$\times P\left[B_j \varphi(k+i|k)\right] + \left[B_j \varphi(k+i|k)\right]^T P\left[(A_j + B_j H)x(k+i|k)\right]$$
$$+ \left[B_j \varphi(k+i|k)\right]^T P\left[B_j \varphi(k+i|k)\right]$$

Applying Lemma 3, the upper bound of the function $h_2(x, \overline{u})$ is:

$$h_2(x, \overline{u}) \le (1 + \alpha_2)\left[(A_j + B_j H)x(k+i|k)\right]^T P\left[(A_j + B_j H)x(k+i|k)\right]^T$$
$$+ \left(1 + \alpha_2^{-1}\right)\left[B_j \varphi(k+i|k)\right]^T P\left[B_j \varphi(k+i|k)\right]$$

There is:

$$h_2(x, \overline{u}) \le (1 + \alpha_2)\left[(A_j + B_j H)x(k+i|k)\right]^T P\left[(A_j + B_j H)x(k+i|k)\right]$$
$$+ \left(1 + \alpha_2^{-1}\right)\mu_1 B_j^T B_j \varphi(k+i|k)^T \varphi(k+i|k)$$

The term $\psi(k+i|k)^T \psi(k+i|k)$ is bounded as:

$$\psi(k+i|k)^T \varphi(k+i|k) \le \epsilon u(k+i|k)^T u(k+i|k)$$

Thus, we obtain:

$$h_2(x, \overline{u}) \le (1 + \alpha_2)\left[(A_j + B_j H)x(k+i|k)\right]^T$$
$$\times P\left[(A_j + B_j H)x(k+i|k)\right] + \epsilon\left(1 + \alpha_2^{-1}\right)\mu_1 B_j^T B_j u(k+i|k)^T u(k+i|k)$$

Moreover, we define the function $h_3(\overline{u})$ as:

$$h_3(\overline{u}) = \overline{u}(k+i|k)^T R \overline{u}(k+i|k)$$
$$= \left[\overline{u}(k+i|k) - u(k+i|k) + u(k+i|k)\right]^T R\left[\overline{u}(k+i|k) - u(k+i|k) + u(k+i|k)\right]$$
$$= \left[\varphi(k+i|k) + u(k+i|k)\right]^T R\left[\varphi(k+i|k) + u(k+i|k)\right]$$
$$= \left[\varphi(k+i|k)\right]^T R\left[\varphi(k+i|k)\right] + \left[\varphi(k+i|k)\right]^T R\left[u(k+i|k)\right]$$
$$+ \left[u(k+i|k)\right]^T R\left[\varphi(k+i|k)\right] + \left[u(k+i|k)\right]^T R\left[u(k+i|k)\right]$$

Applying Lemma 3, the upper bound of the function $h_3(\overline{u})$ is:

$$h_3(\overline{u}) \le (1 + \alpha_3)u(k+i|k)^T R u(k+i|k) + \left(1 + \alpha_3^{-1}\right)\varphi(k+i|k)^T R \varphi(k+i|k)$$

Consider $R \le \lambda_{2,max} I \le \mu_2 I$, where $\lambda_{2,max}$ is the maximum eigenvalue of $R$ and $\mu_2 I$ is the corresponding upper bound. There is:

$$h_3(\overline{u}) \le (1 + \alpha_3)u(k+i|k)^T R u(k+i|k) + \left(1 + \alpha_3^{-1}\right)\mu_2 \psi(k+i|k)^T \psi(k+i|k)$$

The term $\varphi(k+i|k)^T \varphi(k+i|k)$ is bounded as:

$$\varphi(k+i|k)^T \varphi(k+i|k) \le \epsilon u(k+i|k)^T u(k+i|k)$$

There is:

$$h_3(\overline{u}) \leq (1 + \alpha_3)u(k+i|k)^T Ru(k+i|k) + \epsilon\left(1 + \alpha_3^{-1}\right)\mu_2 u(k+i|k)^T u(k+i|k)$$

By replacing $h_2(x, \overline{u})$ and $h_3(\overline{u})$ in the inequality Equation (A1), the following condition holds for all $i > 0$:

$$
\begin{aligned}
&\left(1 + \alpha_1^{-1}\right)\mu_1 x(k+i|k)^T N^T N x(k+i|k) + (1 + \alpha_1)(1 + \alpha_2)\left[(A_j + B_j H)x(k+i|k)\right]^T \\
&\times P\left[(A_j + B_j H)x(k+i|k)\right] + (1 + \alpha_3)u(k+i|k)^T Ru(k+i|k) \\
&+ \epsilon(1 + \alpha_1)\left(1 + \alpha_2^{-1}\right)\mu_1 B_j^T B_j u(k+i|k)^T u(k+i|k)^T - x(k+i|k)^T Px(k+i|k) \\
&+ \epsilon\left(1 + \alpha_3^{-1}\right)\mu_2 u(k+i|k)^T u(k+i|k) + x(k+i|k)^T S(k+i|k) \leq 0
\end{aligned}
\tag{A4}
$$

By replacing $u(k+1|k) = Hx(k+1|k)$ in Equation (A4), we have:

$$
\begin{aligned}
&(1 + \alpha_1)(1 + \alpha_2)\left[(A_j + B_j H)\right]^T P\left[(A_j + B_j H)\right] + \left(1 + \alpha_1^{-1}\right)\mu_1 N^T N \\
&+ \epsilon(1 + \alpha_1)\left(1 + \alpha_2^{-1}\right)\mu_1 B_j^T B_j H^T H - P + S + (1 + \alpha_3)K^T RK + \epsilon\left(1 + \alpha_3^{-1}\right)\mu_2 H^T H \leq 0
\end{aligned}
$$

Pre-multiply and post-multiply by $Q > 0$ and substitute $Q = \gamma P^{-1}$, $Y = HQ$, $\xi_1 = \gamma\mu_1$, and $\xi_2 = \gamma\mu_2$. Applying Schur complements, we obtain the LMI (Equation (27)).

**Remark A1.** *The proof of Theorem 1 is similar to that of Theorem 2. Therefore, we only present the proof of Theorem 2 in this appendix for brevity.*

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
