# Peer review of "A Robust Model Predictive Control for a Photovoltaic Pumping System Subject to Actuator Saturation Nonlinearity"

_sustainability, doi:10.3390/su15054493_

Round 1

Reviewer 1 Report

The authors proposed a simulation study in which a Robust Model Predictive Controller (RMPC) for uncertain nonlinear systems, under actuator saturation, is designed. The theoretical analysis seems consistent and leads to interesting results, but the understandability should be improved.

The mathematical framework presented in section 2 and section 3 from Equation 1 to Equation 24 is not so clear. The authors should be done additional efforts for clearly explain the mathematical formulation. It is very important because the topic is interesting, but in the present form is understandable for reader strictly involved in Robust Model Predictive Controller concepts. In my opinion for a broad approach the introduction section and the abstract should be slightly revised in which the novelty and originality of the study should be better highlighted in the rational use of the solar energy perspective. Please, I suggest introducing a didactic approach for achieving a widest range of readers. This can be useful for appreciate the interesting, proposed research study.

The following recommendations should be carefully addressed before evaluating the manuscript for publication:

1) In the introduction section is well stated:

“Photovoltaic (PV) Energy has gained considerable attraction as a sustainable energy source in the last three decades. The usage of this source of energy has been significantly increased in numerous applications,……”. For this reason, I suggest inserting, inside the body of the Introduction' section, a very short consideration in which is highlighted that the solar energy technology scenario is now quite diversified and exhibits versatility in converting solar radiation into other useful energy forms (for powering extensive applications). About this aspect it may be useful to incorporate the following suitable references.

https://doi.org/10.1016/j.egypro.2015.11.392

https://doi.org/10.1016/j.egypro.2015.11.002

2) At page 7, line 280, close the end of the page, the word “that” appears twice (“that that”), please remove one of them.

3) At page 9, line 301. The order of the quantities explained is wrong. The term “respectively”, doesn’t reflect the explanation that precedes.

4) Page 9, at line 305, the solar irradiation level, the temperature and shadowing conditions are correctly reported as factors that affects the PV conversion efficiency. For completeness, I suggest considering the following additional aspects:

a) non – uniformity of the incident solar flux. Indeed, uneven illumination effect due to reflections from surrounding reflective bodies (seen by the collectors) tends to cause hot spots, current mismatch, and then a slight reduction of the open circuit voltage and a stronger reduction in the fill factor value.

b) increments, fluctuations and uneven distribution of the cell temperature that have a negative impact on the PV conversion mechanism and on the lifespan of the PV panel.

For a detailed analysis of the mentioned aspects (from theoretical and experimental point of view), please consider that can be useful refers to:

https://doi.org/10.1016/j.enconman.2020.113774

5) Page 9, at line 305, please add one of the terms “level” or “intensity” after “solar irradiance”.

6) Manuscript contains no general nomenclature explaining the quantities that appear in the text. Many quantities found are not defined anywhere properly accompanied with their measurement units. This requires action

7) Page 9, at line 310, I suggest moving the word “with” just before equation 26.

8) At page 13 the “Simulation Results” section is introduced. I have not found detailed/general information about which simulation environment or tool has been utilized in the proposed investigation.

Table 1 at page 14 contains errors. The measurement units of the inductance should be expressed in submultiples of Henry [H], while the measurement units of a capacitance should be expressed in submultiples of Farad [F]. Exactly the opposite of what is reported in table 1.

9) Again, in Table 1 at page 14 please replace the terms “inductor” and “capacitor” with “inductance” and “capacitance”, respectively.

10) Page 14, at line 442, please, after the word “are” add the wording “represented by” otherwise replace “are” with the wording “corresponds to”

11) Acronyms should appear at the first time when the word to which it refers appears for the first time. Again, I recommend to not repeat each time the “word” coupled with its acronym, this is redundant, and the sound is not so good.

Overall considerations

d) Abstract should be thoroughly and deeply revised.

f) Conclusions should be listed and summarized and should be stronger and properly focused on the outcome.

Author Response

We would like to thank you and the anonymous reviewers for the constructive comments. Your comments have provided valuable insights to refine paper’s contents and analysis. Responses to reviewer are given bellow

Responses to Reviewer 1’sComments:

Reviewer comments and Suggestions

  • The mathematical framework presented in section 2 and section 3 from Equation 1 to Equation 24 is not so clear. The authors should be done additional efforts for clearly explain the mathematical formulation. It is very important because the topic is interesting, but in the present form is understandable for reader strictly involved in Robust Model Predictive Controller concepts. In my opinion for a broad approach the introduction section and the abstract should be slightly revised in which the novelty and originality of the study should be better highlighted in the rational use of the solar energy perspective. Please, I suggest introducing a didactic approach for achieving a widest range of readers. This can be useful for appreciate the interesting, proposed research study.

Response:

  • First, we the authors, would like to thank you for all of your questions and suggestions.
  • A mathematical formulas have been added in section 3 to give more details concerning the proposed control method
  • we have referred the reader to all references which explains in detail the mathematical formulas of section 2 and section 3
  • Based on your suggestion, the introduction section and the abstract have been revised
  1. Reviewer comment
  • I suggest inserting, inside the body of the Introduction section, a very short consideration in which is highlighted that the solar energy technology scenario is now quite diversified and exhibits versatility in converting solar radiation into other useful energy forms (for powering extensive applications). About this aspect it may be useful to incorporate the following suitable references.
  • https://doi.org/10.1016/j.egypro.2015.11.392
  • https://doi.org/10.1016/j.egypro.2015.11.002

Response:

  • Again, we thank the reviewer for his recommendation. Based on your suggestion, the authors have added in Introduction section :  “Sunlight can be converted into heat (Solar-thermal energy conversion) [1], electricity (solar-Photovoltaic energy conversion) [2], solar fuel (Hydrogen) genered via photocatalytic water splitting [3-4], or sunlight chemicals, such as H2O2 production [5], CO2 reduction[6] and ammonia synthesis [7] ”
  1. Reviewer comment
  • At page 7, line 280, close the end of the page, the word “that” appears twice (“that that”), please remove one of them.

Response:

  • This error has been corrected.
  1. Reviewer comment
  • At page 9, line 301. The order of the quantities explained is wrong. The term “respectively”, doesn’t reflect the explanation that precedes.

Response:

  • We have rewritten the order of the quantities correctly.

  1. Reviewer comment
  • Page 9, at line 305, the solar irradiation level, the temperature and shadowing conditions are correctly reported as factors that affects the PV conversion efficiency. For completeness, I suggest considering the following additional aspects:
  • a) non – uniformity of the incident solar flux. Indeed, uneven illumination effect due to reflections from surrounding reflective bodies (seen by the collectors) tends to cause hot spots, current mismatch, and then a slight reduction of the open circuit voltage and a stronger reduction in the fill factor value.
  • b) increments, fluctuations and uneven distribution of the cell temperature that have a negative impact on the PV conversion mechanism and on the lifespan of the PV panel.
  • For a detailed analysis of the mentioned aspects (from theoretical and experimental point of view), please consider that can be useful refers to:
  • https://doi.org/10.1016/j.enconman.2020.113774

Response:

  • The authors would like to thank the reviewer for his comments.
  • a) We have noted this good point in the future of our research work, we will try to make an analysis on the effect of the non-uniformity of the incident solar flux, in a photovoltaic system.
  • b) The authors have added a Simulations study in section 5.2.5 (Scenario 3, Figure. 11) to illustrate the cell temperature impact
  1. Reviewer comment
  • Page 9, at line 305, please add one of the terms “level” or “intensity” after “solar irradiance”.

Response:

  • We have added the terms “level” after “solar irradiance”
  1. Reviewer comment
  • Manuscript contains no general nomenclature explaining the quantities that appear in the text. Many quantities found are not defined anywhere properly accompanied with their measurement units. This requires action

Response:

  • We have added a section “Appendix B. Nomenclature”, which comprises the Abbreviations and the Symbols with measurement units
  1. Reviewer comment
  • Page 9, at line 310, I suggest moving the word “with” just before equation 26.

Response:

  • Based on your suggestion, we have moved the word “with” just before equation 26
  1. Reviewer comment
  • At page 13 the “Simulation Results” section is introduced. I have not found detailed/general information about which simulation environment or tool has been utilized in the proposed investigation.
  • Table 1 at page 14 contains errors. The measurement units of the inductance should be expressed in submultiples of Henry [H], while the measurement units of a capacitance should be expressed in submultiples of Farad [F]. Exactly the opposite of what is reported in table 1.

Response:

  • We have mentioned the simulation environment used for Simulation Results in the last paragraph of introduction section: “Next, several simulation results are demonstrated under Matlab environment”
  • We have rewritten the measurement units in Table 1 correctly. (Inductance in Henry and capacitance in Farad).
  1. Reviewer comment
  • Again, in Table 1 at page 14 please replace the terms “inductor” and “capacitor” with “inductance” and “capacitance”, respectively.

Response:

  • We have replaced the term “inductor” by “inductance” and “capacitor” by “capacitance”, in Table 1
  1. Reviewer comment
  • Page 14, at line 442, please, after the word “are” add the wording “represented by” otherwise replace “are” with the wording “corresponds to”

Response:

  • We have added the wording “represented by” after the word “are”
  1. Reviewer comment
  • Acronyms should appear at the first time when the word to which it refers appears for the first time. Again, I recommend to not repeat each time the “word” coupled with its acronym, this is redundant, and the sound is not so good.

Response:

  • We have removed the redundancy by using either the “word” or its acronyms

Reviewer 2 Report

The paper proposes a new robust controller methodology based on the Lyapunov stability theorem.  The target application is a pumping system for pv panel applications. The topic of the paper is accurate, the motivation is showed deeply in the beginning of the introduction. The next parts described well the previous works and showed the difference between the previous paper and the current work of the researchers where this paper related on. The next part of the paper is the methodology which shows the related theorem well, the full derivation is placed in the appendix. The applicability of the paper is shown and validated on a small-scale example. I have no major points with this paper.

Author Response

We would like to thank you and the anonymous reviewers for the constructive comments. Your comments have provided valuable insights to refine paper’s contents and analysis. Responses to reviewer are given bellow.

Responses to Reviewer 2’s Comments:

Reviewer comment

  • The paper proposes a new robust controller methodology based on the Lyapunov stability theorem.  The target application is a pumping system for pv panel applications. The topic of the paper is accurate, the motivation is showed deeply in the beginning of the introduction. The next parts described well the previous works and showed the difference between the previous paper and the current work of the researchers where this paper related on. The next part of the paper is the methodology which shows the related theorem well, the full derivation is placed in the appendix. The applicability of the paper is shown and validated on a small-scale example. I have no major points with this paper.

Response:

  • The authors would like to thank the reviewer for his comments.
  • We have added more paragraphs in the abstract, introduction, section 2 and section 3 as well as adding more simulation tests to improve the contribution of our paper.

Reviewer 3 Report

1.    After reading the article, I am not sure whether the article presented to me fits into the thematic scope & aims of the Sustainability journal. In my opinion, it should be sent to journals like: Automatica, IEE Proceedings-Control Theory and Applications, IEEE Transactions on Automatic Control, IET Control Theory and Applications, etc. For this reason, I recommend the rejection of the article, but I would like to emphasize that I appreciate the work of the Authors.

2.    There are minor editing and text formatting errors in the article content. Despite this, I can state that the article is well written.

3.    The article abounds in mathematics, but it hardly touches on the issues related to technology, sustainable development, or the design methods of photovoltaic systems. The manuscript presents a parametric analysis of the proposed RMPC model and, as mentioned above, should be submitted for review in another journal.

Author Response

We would like to thank you and the anonymous reviewers for the constructive comments. Your comments have provided valuable insights to refine paper’s contents and analysis. Responses to reviewer are given bellow.

Responses to Reviewer 3’s Comments:

  1. Reviewer comment
  • After reading the article, I am not sure whether the article presented to me fits into the thematic scope & aims of the Sustainability journal. In my opinion, it should be sent to journals like: AutomaticaIEE Proceedings-Control Theory and ApplicationsIEEE Transactions on Automatic Control, IET Control Theory and Applications,  For this reason, I recommend the rejection of the article, but I would like to emphasize that I appreciate the work of the Authors.

Response:

  • The authors would like to thank the reviewer for his comments.

-     Until this time, over 900 million people in various countries do not have drinkable water available for consumption. Water pumping systems powered by solar-cell generators is one of the most interesting applications for water supply in rural areas that have a substantial amount of insulation and have no access to an electric grid. Through this paper we try to give theoretical solutions for the development of these systems to participate in the development of human beings in these regions. The Simulation Results are currently under experimental stage and in near future we will publish the first results if they are satisfactory.

  1. Reviewer comment
  • There are minor editing and text formatting errors in the article content. Despite this, I can state that the article is well written.

Response:

  • We would like to thank you for your comments. Many errors have been corrected to improve this paper based on reviewer’s constructive comments and Suggestions.
  1. Reviewer comment
  • The article abounds in mathematics, but it hardly touches on the issues related to technology, sustainable development, or the design methods of photovoltaic systems. The manuscript presents a parametric analysis of the proposed RMPC model and, as mentioned above, should be submitted for review in another journal.

Response:

  • We would like to thank you for your comments. In this work we have focused on theoretical controller design and the modeling of the photovoltaic water pumping systems. The Simulation Results obtained are very satisfactory. In the future, we would like to focus on the experimental validation, where more details related to technology will be addressed such as the PV pumping system sizing, the control and data acquisition devices used in the practical implementation.

Reviewer 4 Report

The subject of the paper is relevant and important, however, the manuscript has the scope to improve. 

Major comments: 

1. Instead of keeping the derivations of lemmas in the paper, the authors may want to move the bulk of the computations into the Appendix. This would improve the readability of the paper.

2. The journal caters to interdisciplinary readers and the authors may want to revise the narrative of the paper so that is easier to follow and understand the big picture. 

3. The authors may want to discuss the policy implications of adopting this new model predictive controller design scheme. In addition, the authors may also want to add a section on the costs and benefits associated with implementing this scheme. 

4. The authors may want to add some sensitivity tests for the simulations. 

5. The authors may also want to explain the rationale behind selecting different scenarios for the simulations and the applications in real-world problems.

Author Response

We would like to thank you and the anonymous reviewers for the constructive comments. Your comments have provided valuable insights to refine paper’s contents and analysis. Responses to reviewer are given bellow.

Responses to Reviewer 4’sComments:

  1. Reviewer comment
  • Instead of keeping the derivations of lemmas in the paper, the authors may want to move the bulk of the computations into the Appendix. This would improve the readability of the paper.

Response:

  • First, we the authors, would like to thank you for all of your questions and suggestions. The lemmas in the paper is as brief as possible. The derivations of the lemmas are not kept in the paper. You can notice that we have referred the reader to all references which he will find the derivations of these lemmas.
  1. Reviewer comment
  • The journal caters to interdisciplinary readers and the authors may want to revise the narrative of the paper so that is easier to follow and understand the big picture. 

Response:

  • We would like to thank you for your comments. To make our paper easier to follow and understand, the introduction section and the abstract have been revised, mathematical formulas have been added in to give more details concerning the proposed control method and we have referred the reader to all references which explains in detail the mathematical formulas in our paper.
  1. Reviewer comment
  • The authors may want to discuss the policy implications of adopting this new model predictive controller design scheme. In addition, the authors may also want to add a section on the costs and benefits associated with implementing this scheme. 

Response:

  • Through this paper we try to give theoretical solutions in modeling and control for the development of Water pumping systems powered by solar-cell generators to participate in the development of human beings in in rural areas. This is what we mentioned in introduction section “photovoltaic water pumping systems for irrigation and water supply in remote areas have been widely implemented due to their unique features of ease of installation, environment-friendly, and low maintenance costs [10,11]. … ”, “The objective of the control process is to keep the PVG voltage at MPP voltage in the presence of the PVG dynamic resistance uncertainty and atmospheric conditions change. … ”
  • In the introduction section, there is a short paragraph that talks about benefits associated with implementing this scheme “ The main advantages of the proposed approach, compared with other well-known RMPC techniques, are its ability to consider both actuator saturation and system non-linearity and the reduction of conservativeness by avoiding a large number of inequalities when the actuator saturation constraint is characterized in terms of the convex hull, therefore, the proposed algorithm guarantees lower computation time compared with other methods. … ”
  • In the future, we would like to focus on the experimental validation, where more details on the costs will be given.
  1. Reviewer comment
  • The authors may want to add some sensitivity tests for the simulations. 

Response:

  • We would like to thank you for your comments. Actually the simulations results includes sensitivity tests as follow:
  • 1- In the Scenario 1 (the input saturation effect. figure 9), our results are compared with the RMPC presented in Poursafar et al [55]. You can notice that the RMPC presented in Poursafar et al [55] do not take into account the input saturation, thus making the system unstable.
  • 2- In section 2.4 Uncertainty Polytope Model. We have presented a theoretical method to deal with polytypic uncertain parameters of the system (the dynamic resistance variations in our case). In the Scenario 4 (The dynamic resistance variations effect), figure 12 demonstrate that our proposed RMPC guarantee the stability of the system in presses of dynamic resistance uncertainty.
  1. Reviewer comment
  • The authors may also want to explain the rationale behind selecting different scenarios for the simulations and the applications in real-world problems.

Response:

  • 1- The rationale behind selecting different scenarios for the simulations:
  • For the Scenario 1 (the input saturation effect) we have mentioned in the introduction section that “The dynamic behavior of power converter systems can be described as bilinear model under saturating control signal. The classical control techniques applied to power converters usually do not take into account the input saturation, which can severely degrade the performances of the closed-loop system, thus making the closed-loop system unstable, especially if the converter is subject to large perturbation.”
  • For the Scenario 2 and 3 (solar radiation and temperature variations effect) we have mentioned in the Abstract section that considered PV system have a highly dependent on operation point and climate conditions of solar radiation and temperature: “…The considered system is a combination of a PVG-converter and DC Motor-Pump which possesses nonlinear behavior along with under saturating control signal highly dependent on operation point and climate conditions of solar radiation and temperature. As a result, the control task is complex due to the nonlinearity of the system and its dependence on climate conditions…. ” .We have mentioned in the introduction section that “ …The objective of the control process is to keep the PVG voltage at MPP voltage in the presence of the PVG dynamic resistance uncertainty and atmospheric conditions change… ”
  • For the Scenario 4 (The dynamic resistance variations effect) we have mentioned in the introduction section that”… when a power converter is employed in solar applications, the control system becomes more complex. Many studies have lately demonstrated that PVG characteristics have a substantial effect on the dynamic behavior of power converters [18]. Since the PVG dynamic resistance is both an environmental variable and operating point dependent, major changes in the PV system might compromise its stability [15]. … ”, “ …The objective of the control process is to keep the PVG voltage at MPP voltage in the presence of the PVG dynamic resistance uncertainty … ”
  • 2- The applications in real-world problems:
  • We have mentioned in the introduction section that we can apply our controller to deal with many complex industrial systems that have the input saturation problem and uncertainty:  ” …The proposed method can be applied to deal with complex industrial systems, such as Computer Numerical Control (CNC) machines, Active Magnetic Bearing (AMB), robot manipulators, overhead cranes, and DC-DC power converters. … ”

Round 2

Reviewer 3 Report

1.     In my opinion, the article is well written and can be published in Sustainability after minor revisions.

2.     The Introduction chapter is written clearly and contains references from the literature closely related to the topic discussed. In the final part of the first chapter, the authors present the purpose and scope of the research work carried out and present the organization of the content of the article.

3.     In the part devoted to mathematical models, the Authors introduced several corrections, which contributed to the readability of the Article, e.g. the PVG model has been described in more detail.

4.     Please check the correctness of data and text formatting e.g. the elementary charge value, T_ref (not "ok"), lacking space (line 358) etc. Further in the text, also appears "oK" which is an error.

5.     Appendix B. Nomenclature, similarly to the comments above, it should be corrected in terms of text formatting, e.g. capital letters, units, upper & lower indices etc.

Author Response

We thank again the reviewer for the time they put in reviewing our paper and look forward to meeting your expectations

Responses to Reviewer 1’sComments:

Reviewer comments and Suggestions

Reviewer comment 1, 2 and 3

  • In my opinion, the article is well written and can be published in Sustainability after minor revisions.
  • The Introduction chapter is written clearly and contains references from the literature closely related to the topic discussed. In the final part of the first chapter, the authors present the purpose and scope of the research work carried out and present the organization of the content of the article.
  • In the part devoted to mathematical models, the Authors introduced several corrections, which contributed to the readability of the Article, e.g. the PVG model has been described in more detail.

Responses to reviewer comment 1, 2 and 3

  • The authors would like to thank to thank you for all of your comments

Reviewer comment 4

  • Please check the correctness of data and text formatting e.g. the elementary charge value, T_ref (not "ok"), lacking space (line 358) etc. Further in the text, also appears "oK" which is an error.

 Responses to reviewer comment 4

  • The authors would like to thank the reviewer for his comment. The value of electron charge and the value of Boltzmann constant have been corrected. The correctness of data and text formatting have been checked, we have rewritten with the measurement unit correctly.

Reviewer comment 5

  • Appendix B. Nomenclature, similarly to the comments above, it should be corrected in terms of text formatting, e.g. capital letters, units, upper & lower indices etc.

Responses to reviewer comment 5

  • Appendix B. has been revised and corrected in terms of text formatting.